# Data mining for prothrombin time and international normalized ratio reference intervals in children

**Muhammad Shariq Shaikh**[ORCID]**\*, Sibtain Ahmed**

Department of Pathology and Laboratory Medicine, Aga Khan University, Karachi, Pakistan

\* muhammad.shariq@aku.edu

## Abstract

Reference intervals (RIs) help physicians in differentiating healthy from sick individuals. The prothrombin time (PT) and International normalized ratio (INR) fluctuate in coagulation pathway defects and have interlaboratory variability due to the instrument/reagent used. As direct method is difficult in children, we chose an indirect data mining method for the determining PT/INR RIs. The indirect method overcomes the substantial financial and logistic challenges, and ethical restrictions in children, moreover, allows partitioning in more fine-grained age groups. Prothrombin Time/INR measurements performed in patients aged birth-18 years between January 2013 and December 2020, were retrieved from laboratory management system of the Aga Khan Hospital. Reference intervals were computed using an indirect KOSMIC algorithm. The KOSMIC package function on the assumption that the non-pathologic samples follow a Gaussian distribution (after Box-Cox transformation of the data), following an elaborate statistical process to isolate distribution of physiological samples from mixed dataset. A total of 56,712 and 52,245 values were retrieved for PT and INR respectively. After the exclusion of patients with multiple specimens obtained during the study period, RIs were calculated for 37,356 (PT) and 37,192 (INR) children with stratification into 9 age groups. A comparison of 2.5th and 97.5th percentile results with those of established RIs from SickKids Handbook of Pediatric Thrombosis and Hemostasis demonstrated good agreement in between different age groups. This study supports data mining as an alternate approach for establishing PT/INR RIs, specifically in resource-limited settings. The results obtained are specific to studied population and instrument/reagent used. The study also allows understanding of fluctuations in coagulation pathways with increasing age and hence better clinical decision-making based on PT and INR results.

## Introduction

Laboratory investigations are integral part of patient management in the current era of companion diagnostics and personalized medicine. Reference intervals (RI) are key element of a comprehensive pathology report that allow healthcare providers understand overall health status of an individual. An RI is described as the span of values derived from the distribution of

**Data Availability Statement:** All relevant data are within the manuscript and its Supporting Information files.

**Funding:** The authors received no specific funding for this work.

**Competing interests:** The authors have declared that no competing interests exist.

the results obtained from a sample of the reference population. Ninety-five percent of healthy participants' test results (2.5th and 97.5th percentiles) fall in this range. Laboratory values vary significantly in children as they undergo progressive physiological changes after birth that become even more pronounced around puberty [1]. Therefore, partitioning of paediatric RIs into distinct age groups is fundamental in optimal patient care.

The prothrombin time (PT) and International normalized ratio (INR) assess the integrity of the final common, and extrinsic pathways of the coagulation cascade. These may be prolonged or shortened in various conditions and have significant interlaboratory variability due to the instrument and reagent used [2, 3]. Levels of vitamin K-dependent factors (FII, FVII, FIX and FX) and of FXI and FXII are less than 40% of adult values till the first month of life. Whereas FV, FVIII, FXIII, fibrinogen and Von Willebrand factor levels might be normal or increased. In essence, blood coagulation capacity increases with age due to gradual increase in the plasma concentration of most coagulation factors [4]. Therefore, age-specific values are essential in segregating pathologic results from normal biological changes for making correct medical decisions.

Whenever possible, RIs should be established using direct collection of blood samples from reference population. This is the recommended method as participants are selected using predefined standards. Furthermore, orchestrated steps are followed throughout from handling of samples to statistical calculation and final inference of results. However, due several ethical and practical limitations most laboratories are hesitant in choosing direct method straightaway. At present, most laboratories in Pakistan use RIs adopted from western literature. Pakistan is a developing country and monetary funds are absent for establishing uniform countrywide data. This leaves children at risk of misdiagnosis and mismanagement. Data mining of large data sets has been utilized substantially in healthcare to facilitate healthcare administration, clinical decision making and to promote public health [5, 6]. This alternative approach has also been described in recent literature as a judicious solution for establishing reference intervals [7–9]. Normal biological test results are sequestered from large pool of mixed physiological and pathological results present in laboratory management system. Reference intervals are then generated using complicated algorithms and statistics.

The purpose of this study is to derive PT/INR RIs in Pakistani children, stratified further into narrow age groups. By data mining large laboratory data, indirectly established population-based age-specific RIs will assist more meticulous medical decisions. The algorithm employed was based on estimation from Mixed Distributions using Truncation Points and the Kolmogorov-Smirnov Distance (kosmic), available as an online platform by Zierk J et al. [7].

## Materials and methods

### Study settings and sample handling

This study was conducted at the Section of Hematology and Transfusion Medicine, Department of Pathology and Laboratory Medicine, Aga Khan University Hospital (AKUH), Pakistan after receiving exemption from institutional ethical review board (2022-7534-21514). All data were fully anonymized before accession. Prothrombin Time/INR measurements performed from January 2013 to December 2020 for both admitted and ambulatory patients, aged birth to 18 years, were retrieved from AKUH Clinical Laboratories' laboratory management system. With its central laboratory situated in Karachi, AKU clinical laboratories are a network of 13 stat labs and nearly 300 collection points scattered across Pakistan. Standard operating procedure was strictly followed during all phases of the testing process. Venous blood was collected in tubes with non-activating surface containing 3.2% buffered trisodium citrate anticoagulant. The recommended anticoagulant to blood ratio of 1:9 with a maximum of ± 10% fill

was ensured for each specimen. In case, sample transportation time was more than 8 hours to either central or one of its stat laboratories, platelet-poor plasma was sent in frozen state on dry ice to nearest destination. PT was measured on Sysmex CS Series Blood Coagulation Analyzers (Sysmex Corporation, Kobe, Japan). The system uses clotting method and 3 different levels (1 normal and 2 therapeutic range) commercial control material (Siemens Healthineers Dade™ Ci-Trol™ Coagulation Controls) are used in each 8-hour shift. External quality assurance is ensured by participating in College of American Pathologist surveys twice a year. For calculation of INR, control PT was calculated as a mean value of 20 healthy adults.

## Statistical calculation

The reference intervals were derived for age portioned groups using an indirect algorithm called as KOSMIC, proposed and validated by Zierk et al. [7]. A statistical program which is implemented within a software package to calculate the Box-Cox transformation parameter lambda ($\lambda$), the truncation interval, and the parameters of the Gaussian distribution Mu ($\mu$) and sigma ($\sigma$) was utilized. This program is based on the proposition that proportion of physiological samples in the input dataset can be exhibited using parametric distribution, subsequently a truncation interval T exists within the dataset in which the proportion of abnormal test results is negligible. Moreover, to project the distribution of non-pathological test results, the lower and upper truncation limits, i.e., T1 and T2, were determined using a "Brute Force" approach. Extensive details of the statistical analysis utilized are available from Zierk J et al. [7]. To support our study, RIs obtained were also compared with those from the SickKids Handbook of Pediatric Thrombosis and Hemostasis [10].

## Results

A total of 56,712 specimens for PT and 52,245 specimens for INR were retrieved from LIS during the study period. Excluding patients with more than 1 specimen, RIs were calculated for 37,356 children for PT and 37,192 children for INR. Results were stratified into 9 age groups; number of samples analyzed in each age groups are shown in Table 1. A comparison of our 2.5th and 97.5th percentile results with those of established RIs from SickKids Handbook of Pediatric Thrombosis and Hemostasis demonstrated good agreement in between different age groups (Tables 2 and 3 and Fig 1).

## Discussion

Development of the human coagulation system in infants and during childhood was first described in 1987 and 1992 respectively by Andrew Maureen [11, 12]. A decade later, need for reagent and analyzer combination specific RIs was emphasized [13]. With continuous introduction of better testing platforms combining novel reagents and instruments, accurate RIs for each test are essential. Unless innovative methods that require minimum blood quantity

**Table 1. Number of PT/INR results anlayzed in different age groups.**

| Parameter | Total results retrieved from LIS (n) | Results analyzed* | Age | | | | | | | | |
|---|---|---|---|---|---|---|---|---|---|---|---|
| | | | Birth-4 days | 5–29 days | 30–89 days | 90–179 days | 180–364 days | 1–5 Years | 6–10 years | 11–16 years | 16–18 years |
| PT | 56,712 | 37,356 | 3,315 | 3,046 | 2,050 | 1,449 | 1,952 | 7,732 | 6,919 | 7,247 | 3,846 |
| INR | 52,245 | 37,192 | 3,155 | 3,039 | 2,092 | 1,430 | 1,961 | 7,657 | 6,823 | 7,216 | 3,819 |

* after exclusion of duplicates.

**Table 2. Comparison of currently used and newly established Lower (LRI) and Upper Reference Intervals (URI) for Prothrombin Time (PT).**

| Age | New LRI | New URI | Currently used LRI | Currently used URI |
|---|---|---|---|---|
| birth– 4 days | 10.2 | 18.6 | 10.1 | 15.9 |
| 5–29 days | 9.5 | 15.6 | 10 | 15.3 |
| 30–89 days | 9.7 | 13.1 | 10 | 14.3 |
| 90–179 days | 9.7 | 13.4 | 10 | 14.2 |
| 180–364 days | 9.5 | 12.9 | 10.7 | 13.9 |
| 1–5 years | 9.7 | 12.7 | 10.6 | 11.4 |
| 6–10 years | 9.7 | 12.7 | 10.1 | 12.1 |
| 11–16 years | 9.6 | 12.7 | 10.2 | 12 |
| 16-above | 9.7 | 12.9 | 9.1 | 13.1 |

for reliable measurement of coagulation cascade are available, ethical dilemmas associated with testing children will continue to hinder establishment of age specific RIs using direct approach. Ranking 5[th] amongst all countries, Pakistan population is equivalent to **2.83%** (>228 million) of the total world population [14]. Regrettably, population specific RIs for hematology analytes, established using recommended guidelines and appropriate statistical calculations are unavailable. The situation is even worse for paediatric population. Currently most laboratories in Pakistan have adopted RIs published in textbooks or articles available on internet. This literature mostly belongs to North American or European territories leaving doubts in clinical decision making of indigenous individuals.

Several authors have described successful generation of RIs using indirect approaches for biochemical analytes [8, 9, 15]. However, literature review revealed dearth of big laboratory data analyses for haematology analytes. Moreover, any initial work done focuses more on adult populations. As data mining approach is completely based on retrieving data from laboratory information system accumulated during usual medical management, it overcomes problems associated with direct blood sampling including but not limited to expenses incurred and participant discomfort in terms of venepuncture and recruitment logistics.

Our study is first of its kind in the region, presenting data mining method for PT/INR as a reasonable substitute. To the best of our knowledge and belief, current study is unique in being first from Pakistan, reporting PT/INR RIs in age-portioned groups of children. Since our laboratory receives specimen from entire country, results obtained therefore are representative of general topography of Pakistan. Notably, the indirect method used in the study allowed us to analyze a very large number of PT/INR results (~110,000 in total) in exigent pediatric cohort.

**Table 3. Comparison of currently used and newly established Lower (LRI) and Upper Reference Intervals (URI) for International Normalized Ratio (INR).**

| Age | New LRI | New URI | Currently used LRI | Currently used URI |
|---|---|---|---|---|
| birth– 4 days | 1.02 | 1.73 | 0.9 | 1.6 |
| 5–29 days | 0.89 | 1.59 | 0.9 | 1.6 |
| 30–89 days | 0.88 | 1.44 | 0.9 | 1.6 |
| 90–179 days | 0.8 | 1.52 | 0.8 | 1.2 |
| 180–364 days | 0.42 | 1.58 | 0.8 | 1.2 |
| 1–5 years | 0.88 | 1.12 | 0.8 | 1.2 |
| 6–10 years | 0.95 | 1.27 | 0.8 | 1.2 |
| 11–16 years | 0.94 | 1.29 | 0.8 | 1.2 |
| 16-above | 0.95 | 1.29 | 0.8 | 1.2 |

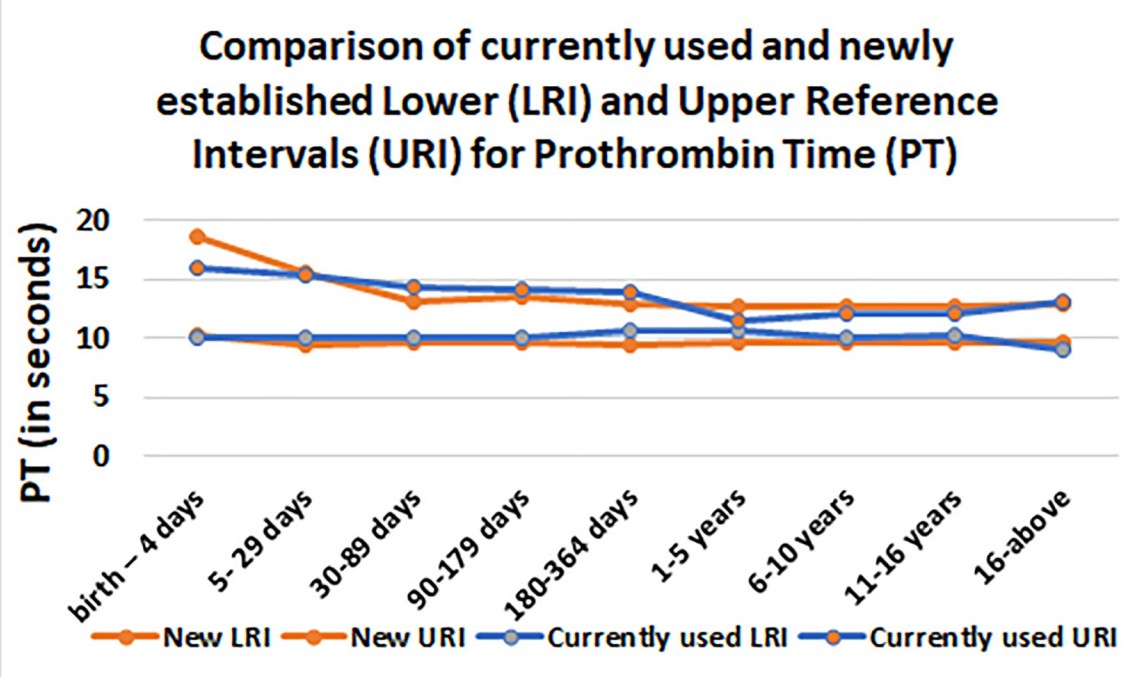

**Fig 1. Comparison of currently used and newly established Lower (LRI) and Upper Reference Intervals (URI) for Prothrombin Time (PT).**

Comparison of our PT/INR results with RIs from the SickKids Handbook of Pediatric Thrombosis and Hemostasis [10] showed high concordance in the reference limits and their age-dependent dynamics. The upper RIs for PT in the birth to 4 days age group are higher compared to the SickKids Cohort, which can be attributed to the population specific inherent biochemical makeup of the Pakistani children evaluated by our study. Secondly the analytical platform utilized can lead to difference in RIs. Moreover, the SickKids cohort was based a priori healthy sample for RI establishment using conventional direct methodology, in contrast. This further supports that RIs established in one population should not be applied to other populations as they can cause misdiagnosis and irreversible and potentially fatal medical consequences.

Our results show prolonged PT/INR levels at birth declining gradually to a steady state after 1 month of age. This finding is in concordance with low levels of vitamin K-dependent factors (FII, FVII, FIX and FX) in first month of life [11]. No difference in values was identified around puberty years.

While, in addition to several merits of indirect approach, any potential differences cannot be analyzed between the groups formulated; hence, individual results have to be complemented with clinical judgement and correlation for optimal inference. Furthermore, a potential limitation of our study is that the statistical web package available calculates only the 2.5th and 97.5% percentile hence the 90% and 95% CI were not reported in this study. Moreover, this study analysed test results from Pakistani children, thus the proposed RIs are directly applicable only for this population and warrants validation before implementation in other population groups.

In conclusion, establishing RIs in children is complex and may be challenging to infer if age-related changes in hemostatic system are not considered. This study supports data mining

as an alternate indirect approach for establishing PT/INR RIs, specifically in resource-limited settings. The results obtained are specific to studied population and instrument/reagent used. It also allows understanding of the fluctuations in extrinsic and common coagulation pathways with increasing age. These intervals therefore, will assist in better clinical decision-making based on PT and INR results.

## Supporting information

**S1 Data.**
(XLSX)

**S2 Data.**
(XLSX)

## Acknowledgments

We wish to acknowledge our IT staff for retrieving the data from Laboratory Information System.

## Author Contributions

**Conceptualization:** Muhammad Shariq Shaikh.

**Data curation:** Muhammad Shariq Shaikh, Sibtain Ahmed.

**Formal analysis:** Sibtain Ahmed.

**Methodology:** Muhammad Shariq Shaikh, Sibtain Ahmed.

**Project administration:** Muhammad Shariq Shaikh.

**Supervision:** Muhammad Shariq Shaikh.

**Writing – original draft:** Muhammad Shariq Shaikh.

**Writing – review & editing:** Muhammad Shariq Shaikh, Sibtain Ahmed.

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
