## [Decision Letter · Decision Letter 0]

31 Aug 2022

PONE-D-22-17903Data mining for prothrombin time and international normalized ratio reference intervals in childrenPLOS ONE

Dear Dr. Shaikh,

Thank you for submitting your manuscript to PLOS ONE. After careful consideration, we feel that it has merit but does not fully meet PLOS ONE’s publication criteria as it currently stands. Therefore, we invite you to submit a revised version of the manuscript that addresses the points raised during the review process.

We look forward to receiving your revised manuscript.

Kind regards,

Zivanai Cuthbert Chapanduka, MBChB (M.D)

Academic Editor

PLOS ONE

Journal Requirements:

Reviewers' comments:

Reviewer's Responses to Questions

**Comments to the Author**

1. Is the manuscript technically sound, and do the data support the conclusions?

Reviewer #1: Yes

Reviewer #2: Yes

2. Has the statistical analysis been performed appropriately and rigorously? 

Reviewer #1: Yes

Reviewer #2: Yes

3. Have the authors made all data underlying the findings in their manuscript fully available?

Reviewer #1: Yes

Reviewer #2: Yes

4. Is the manuscript presented in an intelligible fashion and written in standard English?

Reviewer #1: Yes

Reviewer #2: Yes

5. Review Comments to the Author

Reviewer #1: This is an excellent study, which identified gaps in literature and in the local setting and attempted to address these convincingly.

The abstract does not clearly bring out the aim and the challenges that are then clearly elaborated in the body of the manuscript, that is, no locally established RIs, and ethical challenges to direct approaches to establishing RIs in the population of study.

The methodology in the abstract can be further summarised to accommodate this.

The statistics is quite complex for most of the targeted readers. Zierk J et al did an excellent job in explaining this. It’s not absolutely necessary but would be ideal to explain this with a little more detail.

Reviewer #2: Thank you for the opportunity to review the manuscript entitled " Data mining for prothrombin time and international normalized ratio reference intervals in children".

This research aimed to determine reference intervals (RI) for PT and INR in children using an indirect method, namely the kosmic algorithm. Kosmic uses complex statistics to identify the distribution of non-pathologic samples within a dataset of non-pathological and pathological samples. The authors determined that the RIs for PT established using kosmic at different paediatric ages in Pakastani children were similar to those of a published handbook.

There are some points of concern:

Major points:

1. My main concern is that there is the circular logic in the interpretation of the data. The authors use the kosmic algorithm for calculation of RI in Pakastani children to see whether it is similar or different to published North American/European RIs. Then to validate the RIs obtained by the kosmic algorithm, they make sure the obtained RIs are similar to published reference intervals in the SickKids Handbook (which likely contain North American/European RIs).

Either the study is validating the kosmic algorithm for determination of RIs or the study is determining new RIs for a new population using the established kosmic algorithm, the latter being the stated purpose of the study (line 81) and is supported by the study design.

If there is uncertainty about the validity of kosmic, method validation can be performed by assessing other indirect statistical models (e.g. Hoffman, Bhattacharya, refineR) that can be used to ensure that the values obtained using kosmic are within the 95% CI of the RIs established using kosmic.

It is indeed good practice to compare results obtained in a study with studies that were performed internationally, and I agree that this should be included in the discussion, but this does not validate findings.

2. The 95% CI of the calculated RIs have not been included in this manuscript. While I am not familiar with using kosmic, the 95% CI does appear to be one of the output parameters when using kosmic (DOI: 10.1038/s41598-020-58749-2) and would be welcomed to enhance interpretation of the results of this study.

3. Are there reference intervals for INR in children available in western literature? This can be used for comparison to the INR values obtained in this study, as has been done for PT. If there are not paediatric INR values available in the literature, then the reason for this deserves comment.

4. In the concluding paragraph (line 180-181) and abstract conclusion (line 41-42), it is stated that the results “allow understanding of the fluctuations in extrinsic and common coagulation pathways with increasing age.”

This has not been discussed in relation to the study results within the Discussion. Have the results shown that PT is prolonged at birth and reach a steady state from the age of 1 year? What does this tell us about the extrinsic and common coagulation pathways? Can it be inferred that in children less than 1, these pathways are slower to activate and that this is attributed to the lower levels of vitamin K-dependent factors, as mentioned in the introduction (line 60)?

In addition, there was mention in the Introduction line 55 that “laboratory values may vary…and become more pronounced during puberty”. But there was no comment made about the study results in relation to puberty (or differences between male and female around puberty). In fact, this study showed that the PT around puberty ages are very similar.

Minor points:

- In Abstract, Line 31-33: “Assuming that, non-pathologic samples follow a Gaussian distribution (after Box-Cox transformation of the data), an elaborate statistical process was used to isolate distribution of physiological samples from mixed dataset and used to calculate RIs.”

Is this what the authors performed in addition to what kosmic does, or is that an explanation of how kosmic performs its analysis? It is not clear.

- The SickKids Handbook of Pediatric Thrombosis and Haemostatsis is not referenced.

- Line 85: Zierk J et al is stated as reference 2, but is reference 7.

- Please state in the methods section what was used to calculate INR in the study. In the INR formula, where test PT is divided by mean control PT, is the mean control PT obtained from a calculated mean PT in adult patients?

- The way that the results section is written can be improved. The reference to Table 3 in the text (line 125) is in line with a sentence about comparison with RI from the SickKids Handbook. But Table 3 only contains newly established reference intervals for INR and no comparison to RI from the SickKids Handbook.

- Figure 1: If the URI & LRI for the new RI were in the same colour and the URI & LRI for the currently used RI were in the same colour, this will improve visualisation of data.

- Line 157: “Our study is first of its kind in the region, presenting data mining method as a reasonable substitute.”

Reference 9 performed a similar data mining study in Pakistani children to establish RIs. The author’s research is therefore not the first study using data mining for RI in Pakistan.

6. PLOS authors have the option to publish the peer review history of their article (what does this mean?). If published, this will include your full peer review and any attached files.

Reviewer #1: **Yes: **Leonard Mutema

Reviewer #2: No

---

## [Author Response · Author response to Decision Letter 0]

8 Sep 2022

Point by point answers to reviewer’s queries.

Title: Data mining for prothrombin time and international normalized ratio reference intervals in children

PONE-D-22-17903 

Reviewer 1: This is an excellent study, which identified gaps in literature and in the local setting and attempted to address these convincingly.

The abstract does not clearly bring out the aim and the challenges that are then clearly elaborated in the body of the manuscript, that is, no locally established RIs, and ethical challenges to direct approaches to establishing RIs in the population of study. The methodology in the abstract can be further summarised to accommodate this. The statistics is quite complex for most of the targeted readers. Zierk J et al did an excellent job in explaining this. It’s not absolutely necessary but would be ideal to explain this with a little more detail.

Author’s Response: As per the suggestion, we have elaborated the statistical analysis in the revised manuscript submitted. Moreover, the abstract has also been improved in line with the manuscript. See 'Revised Manuscript with Track Changes' and ‘unmarked revised manuscript.

Reviewer 2: Major points:

1. My main concern is that there is the circular logic in the interpretation of the data. The authors use the kosmic algorithm for calculation of RI in Pakastani children to see whether it is similar or different to published North American/European RIs. Then to validate the RIs obtained by the kosmic algorithm, they make sure the obtained RIs are similar to published reference intervals in the SickKids Handbook (which likely contain North American/European RIs).

Either the study is validating the kosmic algorithm for determination of RIs or the study is determining new RIs for a new population using the established kosmic algorithm, the latter being the stated purpose of the study (line 81) and is supported by the study design.

If there is uncertainty about the validity of kosmic, method validation can be performed by assessing other indirect statistical models (e.g. Hoffman, Bhattacharya, refineR) that can be used to ensure that the values obtained using kosmic are within the 95% CI of the RIs established using kosmic. It is indeed good practice to compare results obtained in a study with studies that were performed internationally, and I agree that this should be included in the discussion, but this does not validate findings.

Author’s Response: For verification of results obtained, we chose SickKids Handbook as a reference because our laboratory was already using SickKids reference intervals in official patient reports. We had to compare new results with already existing and in-practice data. This helped us in identifying any gross change in 2 datasets as well. 

As correctly pointed out by the reviewer, this study is not validating KOSMIC algorithm as it has already been validated by Zierk et al. The study is determining new RIs for a new population using the established kosmic algorithm as stated on line 81 of original submission. 

However, we have modified our text in revised submission in:

- Method section by using: “To support our study” instead of “to validate our study”.

- Discussion section by using: “for optimal inference” instead of “for optimal validation”:

2. The 95% CI of the calculated RIs have not been included in this manuscript. While I am not familiar with using kosmic, the 95% CI does appear to be one of the output parameters when using kosmic (DOI: 10.1038/s41598-020-58749-2) and would be welcomed to enhance interpretation of the results of this study.

Author’s Response: As per the suggestion, the statistical web package available calculates only the 2.5th and 97.5% percentile hence the 90% and 95% CI were not reported in this study. Therefore, it has been included as a limitation under the discussion section. See 'Revised Manuscript with Track Changes' and ‘unmarked revised manuscript.

3. Are there reference intervals for INR in children available in western literature? This can be used for comparison to the INR values obtained in this study, as has been done for PT. If there are not paediatric INR values available in the literature, then the reason for this deserves comment.

Author’s Response: As per the suggestion, reference intervals for INR from western literature (SickKids Handbook) have also been incorporated for comparison. See Table 3.

4. In the concluding paragraph (line 180-181) and abstract conclusion (line 41-42), it is stated that the results “allow understanding of the fluctuations in extrinsic and common coagulation pathways with increasing age.”

This has not been discussed in relation to the study results within the Discussion. Have the results shown that PT is prolonged at birth and reach a steady state from the age of 1 year? What does this tell us about the extrinsic and common coagulation pathways? Can it be inferred that in children less than 1, these pathways are slower to activate and that this is attributed to the lower levels of vitamin K-dependent factors, as mentioned in the introduction (line 60)?

Author’s Response: As per suggestion, following statement has been added in the discussion in relation to the results. 

“Our results show prolonged PT/INR levels at birth declining gradually to a steady state after 1 month of age. This finding is in concordance with low levels of vitamin K-dependent factors (FII, FVII, FIX and FX) in first month of life”. A supporting reference has also been added (Reference no:11).

In addition, there was mention in the Introduction line 55 that “laboratory values may vary…and become more pronounced during puberty”. But there was no comment made about the study results in relation to puberty (or differences between male and female around puberty). In fact, this study showed that the PT around puberty ages are very similar.

Author’s Response: As per the suggestion, following statement has been added in the discussion in relation to results:

“No difference in values was identified around puberty years”.

Minor points:

- In Abstract, Line 31-33: “Assuming that, non-pathologic samples follow a Gaussian distribution (after Box-Cox transformation of the data), an elaborate statistical process was used to isolate distribution of physiological samples from mixed dataset and used to calculate RIs.”

Is this what the authors performed in addition to what kosmic does, or is that an explanation of how kosmic performs its analysis? It is not clear.

Author’s Response: This is an explanation of the functions of KOSMIC, available as a software package. We have elaborated this in the abstract in the revised manuscript submitted.

The SickKids Handbook of Pediatric Thrombosis and Haemostatsis is not referenced.

Author’s Response: The SickKids Handbook of Pediatric Thrombosis and Haemostatsis is now referenced: Reference number: 10

Line 85: Zierk J et al is stated as reference 2, but is reference 7.

Author’s Response: This has been corrected in revised manuscript.

Please state in the methods section what was used to calculate INR in the study. In the INR formula, where test PT is divided by mean control PT, is the mean control PT obtained from a calculated mean PT in adult patients?

Author’s Response: Following statement has been added in method section:

“For calculation of INR, control PT was calculated as a mean value of 20 healthy adults”.

The way that the results section is written can be improved. The reference to Table 3 in the text (line 125) is in line with a sentence about comparison with RI from the SickKids Handbook. But Table 3 only contains newly established reference intervals for INR and no comparison to RI from the SickKids Handbook.

Author’s Response: Results section has now been improved. As INR values (from SickKids Handbook) have now been incorporated for comparison, the reference to Table 3 in the text (line 125 in original submission) is now in line with a sentence about comparison with RI from the SickKids Handbook.

Figure 1: If the URI & LRI for the new RI were in the same colour and the URI & LRI for the currently used RI were in the same colour, this will improve visualisation of data.

Author’s Response: Colours in Figure 1 have been changed according to reviewer’s suggestion.

Line 157: “Our study is first of its kind in the region, presenting data mining method as a reasonable substitute.” Reference 9 performed a similar data mining study in Pakistani children to establish RIs. The author’s research is therefore not the first study using data mining for RI in Pakistan.

Author’s Response: Reference 9 study was done for “Alkaline Phosphatase levels”. As far as thrombosis and hemostasis are concerned, our study is first of its kind in the region reporting Prothrombin Time/INR RIs by data mining. However, we have now specifically mentioned PT/INR in the same sentence for clarification. The corrected statement now reads as follows:

“Our study is first of its kind in the region, presenting data mining method for PT/INR as a reasonable substitute”.

---

## [Decision Letter · Decision Letter 1]

14 Oct 2022

Data mining for prothrombin time and international normalized ratio reference intervals in children

PONE-D-22-17903R1

Dear Dr. Muhammad Shariq Shaikh

We’re pleased to inform you that your manuscript has been judged scientifically suitable for publication and will be formally accepted for publication once it meets all outstanding technical requirements.

Kind regards,

Zivanai Cuthbert Chapanduka, MBChB (M.D)

Academic Editor

PLOS ONE

Additional Editor Comments (optional):

Reviewers' comments:

Reviewer's Responses to Questions

**Comments to the Author**

1. If the authors have adequately addressed your comments raised in a previous round of review and you feel that this manuscript is now acceptable for publication, you may indicate that here to bypass the “Comments to the Author” section, enter your conflict of interest statement in the “Confidential to Editor” section, and submit your "Accept" recommendation.

Reviewer #1: All comments have been addressed

Reviewer #2: All comments have been addressed

2. Is the manuscript technically sound, and do the data support the conclusions?

Reviewer #1: Yes

Reviewer #2: Yes

3. Has the statistical analysis been performed appropriately and rigorously? 

Reviewer #1: (No Response)

Reviewer #2: Yes

4. Have the authors made all data underlying the findings in their manuscript fully available?

Reviewer #1: Yes

Reviewer #2: Yes

5. Is the manuscript presented in an intelligible fashion and written in standard English?

Reviewer #1: Yes

Reviewer #2: Yes

6. Review Comments to the Author

Reviewer #1: (No Response)

Reviewer #2: (No Response)

7. PLOS authors have the option to publish the peer review history of their article (what does this mean?). If published, this will include your full peer review and any attached files.

Reviewer #1: **Yes: **Leonard Mutema

Reviewer #2: No

---

## [Editor Report · Acceptance letter]

20 Oct 2022

PONE-D-22-17903R1 

Data mining for prothrombin time and international normalized ratio reference intervals in children 

Dear Dr. Shaikh:

I'm pleased to inform you that your manuscript has been deemed suitable for publication in PLOS ONE. Congratulations! Your manuscript is now with our production department. 

Kind regards, 

on behalf of

Dr. Zivanai Cuthbert Chapanduka 

Academic Editor

PLOS ONE